# Sex Disparity in Bilateral Asymmetry of Impact Forces during Height-Adjusted Drop Jumps

**DOI:** 10.3390/ijerph18115953

**Published:** 2021-06-01

**Authors:** Chin-Yi Gu, Xiang-Rui Li, Chien-Ting Lai, Jin-Jiang Gao, I-Lin Wang, Li-I Wang

**Affiliations:** 1Department of Education and Human Potentials Development, National Dong Hwa University, No. 1, Sec. 2, Da Hsueh Rd., Shoufeng, Hualien 97401, Taiwan; stronggu@gmail.com; 2Department of Physical Education and Kinesiology, National Dong Hwa University, No. 1, Sec. 2, Da Hsueh Rd., Shoufeng, Hualien 97401, Taiwan; s86919177@gmail.com (X.-R.L.); 610789004@gms.ndhu.edu.tw (C.-T.L.); kao974129@gmail.com (J.-J.G.); 3College of Physical Education, Hubei Normal University, Huangshi 435002, China

**Keywords:** anterior cruciate ligament, ground reaction force, knee injuries, leg dominance

## Abstract

Side-to-side asymmetry of lower extremities may influence the risk of injury associated with drop jump. Moreover, drop heights using relative height across individuals based on respective jumping abilities could better explain lower-extremity loading impact for different genders. The purpose of the current study was to evaluate the sex differences of impact forces and asymmetry during the landing phase of drop-jump tasks using drop heights, set according to participants’ maximum jumping height. Ten male and ten female athletes performed drop-jump tasks on two force plates, and ground reaction force data were collected. Both feet needed to land entirely on the dedicated force plates as simultaneously as possible. Ground reaction forces and asymmetry between legs were calculated for jumps from 100%, 130%, and 160% of each participant’s maximum jumping height. Females landed with greater asymmetry at time of contact initiation and time of peak impact force and had more asymmetrical peak impact force than males. Greater values and shorter time after ground contact of peak impact force were found when the drop height increased to 160% of maximum jumping ability as compared to 100% and 130%. Females exhibited greater asymmetry than males during drop jumps from relative heights, which may relate to the higher risk of anterior cruciate ligament injury among females. Greater sex disparity was evident in impact force asymmetry than in the magnitude of peak impact force; therefore, it may be a more appropriate field-screening test for risk of anterior cruciate ligament injury.

## 1. Introduction

Compared to males, females have a high rate of noncontact anterior cruciate ligament (ACL) injuries during athletic competition [1]. Many studies have reported that females showed greater impact in vertical ground reaction forces (I-vGRF) and have a strong trend toward one-leg dominance with greater bilateral asymmetry during landing tasks compared to males [2,3,4]. Previous research has revealed that females have larger I-vGRF in the nondominant limb compared to that in the dominant limb during the landing phase of the drop-jump (DJ) task [5]. Greater I-vGRF can create knee instability and increase loading both on the knee joint and ACL [6]. Leg dominance theories may explain reports that side-to-side imbalances predict future ACL injury risk during bilateral tasks [2], and side-to-side asymmetries of I-vGRFs during bilateral landing tasks may be an important predictor for ACL injury risk [7]. Despite the many studies that have focused on asymmetry of the lower extremity during landing in females, little research has looked specifically at I-vGRF parameters to measure asymmetry during bilateral landing tasks between sexes. 

Drop jumps (DJs), a popular plyometric exercise method for improving the neuromuscular strength and power of lower extremities, are widely used in biomechanical studies to investigate lower-extremity injury risk [8,9,10]. The DJ involves stepping off a fixed drop height, briefly landing, and immediately taking off from the ground to achieve maximum jump height. Previous studies have investigated DJ performance under various height and loading parameters and indicated that the choice of appropriate drop height is important [9,11]. Some studies suggested that during double-legged DJ training, the drop height should not exceed 40 cm [8,10]. Some studies indicated that bilateral DJs from lower heights, such as 20 cm, caused greater asymmetry in leg power required, which may cause an asymmetrical bilateral training stimulus. [12].

Previous studies often investigated the sex disparities in injury risk in controlled laboratory experiments using the same drop heights (absolute height) in males and females [13,14]. However, Komi and Bosco have suggested that the appropriate DJ height differs between males and females [15], and that the capacity to endure the impact forces is weaker for females, both of which jointly contributed to the sex disparity in the stiffness alterations when the drop height increased to 60 cm [14]. Therefore, the most appropriate drop heights for plyometric training may vary across individuals according to jumping ability [16]. To our knowledge, this study is among the first to investigate sex disparity in bilateral asymmetry using drop jumps from heights set relative to individuals’ jump ability.

The few studies comparing the I-vGRF of males and females dropping from relative height (RH) have found that there are no sex differences during unilateral landings [17,18]. However, these studies were only focused on one RH (100% jump height of the countermovement jump). Increased drop height during bilateral DJ would generate greater perturbation of landing and likely affect the landing impact and bilateral asymmetry [7,10]. The purpose of the current study was to examine the differences in bilateral landings between male and female athletes performing two-legged DJs from RHs. We hypothesized that: (a) there would be no sex differences in landing impact; (b) females would exhibit greater bilateral asymmetry compared to males when performing DJs from RHs. We also aimed to explore the impact force of incremental drop height changes on the DJ for determining an appropriate RH.

## 2. Materials and Methods

### 2.1. Participants

Ten male (1.75 ± 0.07 m and 69.9 ± 8.9 kg) and ten female (1.61 ± 0.04 m and 52.4 ± 4.2 kg) college athletes from the physical education department participated, and their written informed consent was collected. Males included 7 basketball, 2 track and field, and 1 field hockey athletes. Females included 5 basketball, 4 track and field, and 1 field hockey stick athletes. All participants had plyometric training experience and were familiar with double-legged drop jumps. Participants had been free of any lower-extremity injury in the past 6 months prior to the experiment. The study met the principles outlined in the Declaration of Helsinki [19]. 

### 2.2. Countermovement Jump Testing Procedures

For all testing, the participants warmed up by running 10 min at a comfortable self-selected pace on a motorized treadmill and dynamically stretching the lower extremity prior to the testing protocol. At least one week before DJ experimentation, the participants’ jumping ability was measured using the countermovement jump task. After the warm-up, there was a 3 min rest. Participants were then instructed to stand erect on a force plate (BP600900, AMTI Inc., Watertown, MA, USA) and with their hands placed on their waist. Three maximum-effort vertical countermovement jumps were performed with a 1 min rest between trials, and ground reaction forces (GRFs) were recorded at 2000 Hz with KwonGRF software (VISOL Inc., Seoul, Korea). Jump height (JH) was calculated using the formula: JH = gT^2^/8 (g = 9.81 m/s^2^; T = flight time) [9]. The greatest JH was used to determine each participant’s RHs for the subsequent DJ experimental session.

### 2.3. Drop-Jump Testing Procedures

Three minutes after the warm-up, the participants completed 3–4 practice jumps to become familiar with the appropriate DJ technique under the guidance of the Strength and Conditioning Coach. The drop-jump tasks consisted of three RHs (JH × 100% (males: 46.70 ± 3.90 cm; females: 31.30 ± 2.24 cm), JH × 130% (males: 60.71 ± 5.34 cm; females: 40.69 ± 3.07 cm), and JH × 160% (males: 74.72 ± 6.58 cm; females: 50.08 ± 3.78 cm) (RH_100%_, RH_130%_, and RH_160%_, respectively)), and the trial height order was randomized for each participant. The double-legged DJ tasks comprised dropping with both feet off a raised platform onto two force plates and then jumping off the ground as fast and high as possible. Participants were instructed to keep their hands on their waist. During the DJ task instructions, the participants were asked to control their legs in the same way. Each entire foot needed to land on a separate force plate at the same time as possible. Each participant was asked to complete three successful DJs (as described previously) for each RH with a 30 s rest between trials. Data were averaged across the three trials for each DJ.

### 2.4. Data Collection and Analysis

The GRF data were collected and analyzed with the Qualisys Track Manager motion capture and analog data acquisition system (Qualisys, Gothenburg, Sweden). Two force plates (BP600900; AMTI, Inc., Watertown, MA, USA) at a 2000 Hz sampling rate were used. The force plates were synchronized using a Qualisys 64-channel A/D board. The ground contact phase was defined as the time interval from foot contact to the foot leaving the ground. Parameters were calculated for the participants’ dominant and nondominant limbs. The dominant limb was determined using a ball-kicking test. The leg they chose to kick with was deemed the dominant leg, whereas the plant leg was deemed the nondominant leg [20]. The instant of ground contact commencement (t1) was determined by assessing the 10 N vGRF threshold of the force plate. I-vGRFpeak was defined as the maximum vGRF during the landing phase of the DJ. Time to I-vGRFpeak after ground contact was considered as the time interval from t1 to the time of I-vGRFpeak (t2): t2-t1. Absolute difference in t1, t2, and I-vGRFpeak per leg during the landing phase was used to calculate side-to-side asymmetry [7,12,21]. The I-vGRFs were normalized using two methods: (a) normalized to body weight (BW) [7], and (b) normalized to body weight × RH^−½^ (BW × RH^−½^) to minimize the variation due to individual differences in body weight and drop height [17]. 

### 2.5. Statistical Analysis

Data were analyzed using the Statistical Program for the Social Sciences 14.0 for Windows (SPSS, Inc., Chicago, IL, USA). Variables were analyzed with a two-way mixed-model ANOVA for sex and three RHs. If statistically significant interactions existed, a post-hoc using independent *t*-test for sex difference and one-way ANOVA for each RH were performed. If no statistically significant interaction existed, main effects between sexes or among RHs were analyzed. If a difference of main effects among RHs existed, the post-hoc test was assessed using the least significant difference (LSD) method of variances. Sphericity was analyzed using Mauchly’s test, followed by a Greenhouse–Geisser correction when the results were nonspherical. The significance level was set at α = 0.05. A priori power analysis (G × Power version 3.1.9.4; Heinrich Heine University Düsseldorf, Düsseldorf, Germany), with a power level of 80% and an α level of 0.05 [22], was performed. The expected effect size was calculated using the means (2.16 and 2.98) and standard deviation (0.46 and 0.78) of the peak impact force under DJH30 and DJH40 conditions [23]. It revealed that the sample size of 14 participants would be sufficient for the analysis.

## 3. Results

The time to peak I-vGRFpeak variables are shown in Table 1. No interaction between height and sex was observed in time to I-vGRFpeak after ground contact (dominant limb: *p* = 0.238; nondominant limb: *p* = 0.203). A significant main effect of height was observed (dominant limb: ES 0.8, *p* = 0.024; nondominant limb: ES 0.1, *p* = 0.009). Times to I-vGRFpeak after ground contact in RH_160%_ were significantly shorter than in RH_100%_ (dominant limb: *p* = 0.021; nondominant limb: *p* = 0.013) and RH_130%_ (dominant limb: *p* < 0.001; nondominant limb: *p* = 0.012). No difference was found between RH_100%_ and RH_130%_ (dominant limb: *p* = 0.925; nondominant limb: *p* = 0.467). No significant main effect of sex was found (dominant limb: *p* = 0.124; nondominant limb: *p* = 0.154). 

The I-vGRFpeak variables are shown in Table 2. When the I-vGRFpeak was normalized by BW, no interaction between height and sex was observed (dominant limb: *p* = 0.541; nondominant limb: *p* = 0.144). A significant main effect of height was observed (dominant limb: ES 0.25, *p* < 0.001; nondominant limb: ES 0.30, *p* < 0.001). The I-vGRFpeak significantly increased with each drop height increment in the dominant limb (RH_100%_ and RH_130%_: *p* < 0.001; RH_100%_ and RH_160%_: *p* < 0.001; RH_130%_ and RH_160%_: *p* < 0.001) and in the nondominant limb (RH_100%_ and RH_130%_: *p* < 0.001; RH_100%_ and RH_160%_: *p* < 0.001; RH_130%_ and RH_160%_: *p* < 0.001). The I-vGRFpeak of males was significantly greater than females (dominant limb: *p* = 0.007; nondominant limb: *p* = 0.013). When the I-vGRFpeak was normalized by BW × RH^−½^, no interaction between height and sex was observed (dominant limb: *p* = 0.853; nondominant limb: *p* = 0.513), but a significant main effect of height was observed (dominant limb: ES 0.08, *p* = 0.001; nondominant limb: ES 0.05, *p* = 0.003). The I-vGRFpeak of RH_160%_ was significantly greater than of RH_100%_ in both limbs (dominant limb: *p* = 0.006; nondominant limb: *p* = 0.003) and RH_130%_ in dominant limb (*p* = 0.017). No difference was found between RH_100%_ and RH_130%_ in both limbs dominant limb: *p* = 0.083; nondominant limb: *p* = 0.086) and between RH_130%_ and RH_160%_ in the nondominant limb (*p* = 0.061). No significant main effect was found between sexes (dominant limb: *p* = 0.493; nondominant limb: *p* = 0.524). 

The absolute differentials in time of I-vGRFpeak between the dominant and nondominant limbs are shown in Figure 1. No interaction between height and sex was observed in the absolute time differentials of the I-vGRFpeak (*p* = 0.184). No significant main effect of height variables was observed (*p* = 0.084). The absolute time differentials in the I-vGRFpeak of females were significantly greater than males (ES 0.20, *p* = 0.010). 

The absolute differentials in starting contacts between the dominant and nondominant limbs are shown in Figure 2. Significant effects were not found for the interaction of height and sex on the absolute differentials in time of starting dominant limb and nondominant limb contacts (*p* = 0.333). No significant main effect of height variables was observed (*p* = 0.918). The absolute time differentials of starting contacts of females were significantly greater than males (ES 0.25, *p* = 0.007). 

The absolute differences in I-vGRFpeak between the dominant and nondominant limbs are shown in Table 3. No interaction between height and sex was observed when normalized by BW (*p* = 0.926) or by BW × RH^−½^ (*p* = 0.773). No significant main effect of height was observed (BW: *p* = 0.110; BW × RH^−½^: *p* = 0.472). The absolute differentials of females were significantly greater compared to those of males (BW: *p* = 0.014; BW × RH^−½^: *p* = 0.002).

## 4. Discussion

The aim of this study was to compare the I-vGRFs of bilateral legs between males and females to assess the symmetry of impact induced by DJ from different RHs according to an individuals’ jump ability. Females and males were found to have a similar I-vGRFpeak overall. However, females had greater asymmetry in starting contact times, time of I-vGRFpeak, and I-vGRFpeak than males. A greater I-vGRFpeak and shorter time to I-vGRFpeak after ground contact was found at RH_160%_ than RH_100%_ and RH_130%_ overall. 

Previous studies have indicated that the risk of lower-extremity injuries may be associated with the magnitude of peak impact and time between peak impact and ground contact [9,10]. Our study showed that the time to I-vGRFpeak during DJ was earlier at RH_160%_ than RH_100%_ and RH_130%_, and I-vGRFpeak was increased with incremental RH, which could increase the risk of injury [6,24]. This study followed the normalized suggestion by [17] to minimize the variation due to individual differences in body weight and drop height. However, the impact force was still significantly greater in RH_160%_ than RH_100%_ and RH_130%_ after normalization, which suggests that the increase in impact force at RH_160%_ is not only affected by the increase in drop height. These data suggest that a relatively high impact load of the lower extremity may be linked to practices of the double-legged DJ from a height above 160% height of the countermovement jump. 

It is important to note that the height of the countermovement jump for males is nearly 15 cm higher than that for females; this disparity implies that males have greater muscle performance in the lower extremity [25]. Therefore, it is necessary to consider that DJs of absolute height may create an inequitable task demand and unrealistic impact on the lower extremities of female participants [17]. The present study found that peak I-vGRFs were greater in males than females during DJs from RHs. As previously described, greater drop heights prior to landing incrementally increase peak I-vGRFs on the lower extremity. The higher drop height in males in our study was set relative to the higher jump height of the countermovement jump, which may have caused the contrary finding with previous research on the experimental design of absolute drop height [4] but may justify the greater peak I-vGRFs recorded in males. Interestingly, while these peak I-vGRFs were normalized by the square root of drop height, the sex disparities disappeared. Wang et al. reported that the capacity to dissipate or endure the impact forces experienced by the lower extremity might be stronger in males compared with that of their female counterparts [14]. Therefore, it remains to be clarified whether males will have a greater injury risk when the DJ is adjusted for RH. Regardless of which normalization method is selected, females do not experience greater impact than males when the drop height of the DJ task is set relative to jumping ability. 

A previous study found females to have greater side-to-side asymmetries than males during landing tasks, but the measurement was limited to kinematic and kinetic parameters from the absolute drop height [3]. To the best of our knowledge, ours is the first study to demonstrate that side-to-side asymmetries in the I-vGRFpeak parameters during bilateral RH DJ tasks differ between males and females. The present study found that females exhibit more asymmetrical magnitude and timing in the I-vGRFpeak than males during RH DJ. The greater imbalance of the I-vGRFpeak supports the idea that females may adopt a leg-laterality strategy during landing, relying on laterality of the limb to resist and absorb GRFs and possibly increasing the risk of knee injury [5,17]. A greater asymmetrical I-vGRFpeak between sides may generate greater knee instability and a high risk of second ACL injury [11,26,27]. Variation within side-to-side asymmetry in the I-vGRFpeak may provide clinicians and scientists with a method to identify athletes predisposed to potential future knee injury [7], especially females, whose greater asymmetry of impact may explain their greater risk of ACL injury. In the current study, male and female subjects used the same instructions of the DJ task and had similar sports experiences. Among them, athletes of the ipsilateral sports, such as badminton, table tennis, and tennis, were not included. Therefore, we infer that female had larger bilateral asymmetry, which may be due to an individual’s undesirable neuromuscular control.

The current study found a significantly greater time difference between starting dominant and nondominant limb contacts in females relative to males. The bilateral differentials in the timing of contact initiation could cause an asymmetrical bilateral stimulus and will affect the neuromuscular load between legs [21,28,29]. Ball et al. attributed the asymmetry in contact initiation to an insufficient drop time from lower heights [12]. Because of their lower average jump heights, females had lower drop heights in our study, but there was no difference found in the side-to-side asymmetry of starting contact times at different heights. The difference in the findings of the two studies may be explained by the different methodology, particularly the different platform-leaving movement: we asked participants to leave the raised platform symmetrically, while the previous participants were asked to leap off the platform with the right foot leading and needed to bring their trailing foot into position to become bilaterally coordinated before landing [12]. This indicates that the females DJs from lower heights with shorter drop times should not be the cause of greater asymmetry of starting contact times. This result showed that females might be more prone to preferential use of one leg over the other in DJ tasks [30]. 

This study had several limitations. First, joint movement and internal kinetics of the lower extremity were not measured. Therefore, future studies should use kinematical analysis and the inverse dynamic method to confirm the method by which I-vGRF side-to-side imbalances may cause ACL injury. Second, the sample of this study was also healthy and without pathology, and the cross-sectional design cannot assess injury risk. Otherwise, we assume, based on previous studies, that a greater jumping height is relative to high strength levels of the lower extremity. Therefore, the RHs used in this study were according to individual jumping ability instead of controlling for strength levels. We must also be aware of the inconsistency of statistical results found between different normalization methods of I-vGRFpeak; the question of whether males will exhibit greater impact load during DJs from RHs is still unanswered.

## 5. Conclusions

According to the results of this study, the use of relative height may be a better choice rather than absolute height, and it is not recommended that athletes practice double-legged drop jumps from heights above 160% of maximum jump height. In addition, athletes must pay attention to the bilateral asymmetries in the risk of lower limb injury during the drop-jump task, especially female athletes. Impact force asymmetry may be a better screening tool for lower-extremity injury risk than magnitude. The impact force of asymmetry may serve as useful monitoring information for practitioners.

## Figures and Tables

**Figure 1 ijerph-18-05953-f001:**
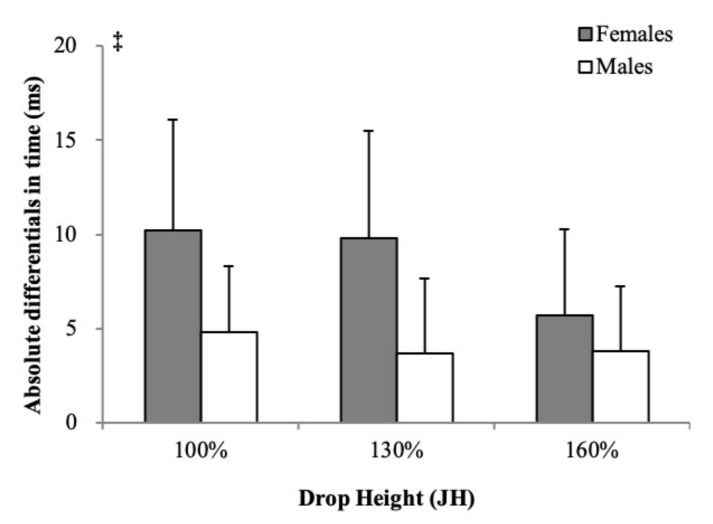
The side-to-side differentials in time of I-vGRFpeak between the dominant and nondominant limbs. I-vGRFpeak = peak impact force during the landing phase; JH = jump height of the countermovement jump. ‡ Significant sex main effect (*p* < 0.05).

**Figure 2 ijerph-18-05953-f002:**
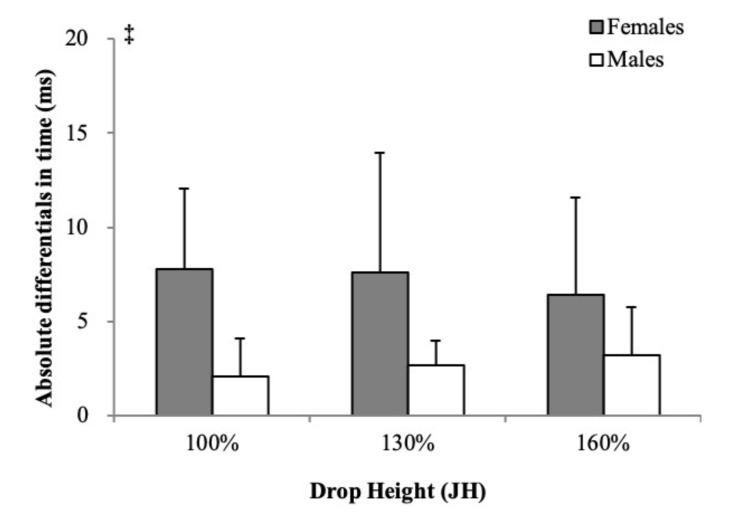
The side-to-side differentials in starting contacts between the dominant and nondominant limbs. JH = jump height of the countermovement jump. ‡ Significant sex main effect (*p* < 0.05).

**Table 1 ijerph-18-05953-t001:** Time to I-vGRFpeak after ground contact for both limbs.

	RH_100%_	RH_130%_	RH_160%_	RH × sex	RH	Sex
**Dominant limb (ms) ^†^**						
Females	47.60 (10.43)	50.60 (7.60)	43.60 (4.40)	*p* = 0.238	*p* = 0.024	*p* = 0.124
Males	45.70 (7.32)	43.10 (6.85)	40.50 (5.42)			
*Post Hoc*	RH_100%_, RH_130%_ > RH_160%_					
**Nondominant limb (ms)** ^†^						
Females	47.10 (10.14)	48.80 (7.64)	44.10 (5.17)	*p* = 0.203	*p* = 0.009	*p* = 0.154
Males	47.10 (6.40)	42.90 (6.42)	39.50 (3.31)			
*Post Hoc*	RH_100%_, RH_130%_ > RH_160%_					

Values are presented with mean and standard error of the mean. I-vGRFpeak = peak impact force during the landing phase; RH = relative height. ^†^ Significant RH main effect (*p* < 0.05).

**Table 2 ijerph-18-05953-t002:** I-vGRFpeak for both limbs.

	RH_100%_	RH_130%_	RH_160%_	RH × sex	RH	Sex
**I-vGRFpeak (BW)**						
**Dominant limb** ^†^^,^^‡^						
Females	1.63 (0.53)	1.98 (0.49)	2.44 (0.40)	*p* = 0.541	*p* < 0.001	*p* = 0.007
Males	2.11 (0.36)	2.64 (0.52)	3.10 (0.68)			
*Post Hoc*	RH_100%_ < RH_130%_ < RH_160%_; Females < Males					
**Nondominant limb** ^†, ‡^						
Females	1.62(0.29)	1.91(0.52)	2.27(0.58)	*p* = 0.144	*p* < 0.001	*p* = 0.013
Males	2.01 (0.45)	2.55 (0.59)	3.02 (0.69)			
*Post Hoc*	RH_100%_ < RH_130%_ < RH_160%_; Females < Males					
**I-vGRFpeak (BW×RH^−½^)**						
**Dominant limb** ^†^						
Females	2.92(0.96)	3.11(0.78)	3.45(0.61)	*p* = 0.853	*p* = 0.001	*p* = 0.493
Males	3.10(0.57)	3.40 (0.69)	3.60 (0.81)			
*Post Hoc*	RH_100%_ = RH_130%_ < RH_160%_					
**Nondominant limb** ^†^						
Females	2.91(0.61)	3.00(0.82)	3.21(0.85)	*p* = 0.513	*p* = 0.003	*p* = 0.524
Males	2.96 (0.68)	3.28 (0.81)	3.50 (0.81)			
*Post Hoc*	RH_100%_ < RH_160%_					

Values are presented with mean and standard error of the mean. I-vGRFpeak = peak impact force during the landing phase; BW = body weight; RH = relative height. ^†^ Significant RH main effect (*p* < 0.05), ^‡^ significant sex main effect (*p* < 0.05).

**Table 3 ijerph-18-05953-t003:** Side-to-side differences of the I-vGRFpeak.

	RH_100%_	RH_130%_	RH_160%_	RH × sex	RH	Sex
**I-vGRFpeak (BW)** ^‡^						
Females	0.46 (0.28)	0.58 (0.26)	0.54 (0.31)	*p* = 0.926	*p* = 0.110	*p* = 0.014
Males	0.25 (0.08)	0.37 (0.20)	0.37 (0.12)			
*Post Hoc*	Males < Females					
**I-vGRFpeak (BW** **×** **RH^−½^)** ^‡^						
Females	0.82 (0.49)	0.90 (0.39)	0.76 (0.43)	*p* = 0.773	*p* = 0.472	*p* = 0.002
Males	0.37 (0.12)	0.48 (0.27)	0.43 (0.14)			
*Post Hoc*	Males < Females					

Values are presented with mean and standard error of the mean. I-vGRFpeak = peak impact force during the landing phase; BW = body weight; RH = relative height. ^‡^ Significant sex main effect (*p* < 0.05).

## Data Availability

The data used to support the findings of this study are available from the corresponding author upon request.

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
