# Peer review of "Sex Disparity in Bilateral Asymmetry of Impact Forces during Height-Adjusted Drop Jumps"

_ijerph, 2021, doi:10.3390/ijerph18115953_

Round 1
Reviewer 1 Report
1) The statistical method used in this study needs to be modified.
If you want to compare gender and each DJ in this study, please use two-way means ANOVA.
If there is an interaction, please conduct a post-examination using independent t-test for gender difference and one-way ANOVA for each DJ.
2) The tables and figures presented in this study have difficulties in interpreting the results. For the convenience of readers, please make it easier to correct the categories presented and indicate the data, statistical values, and p values.
3) There is no indication of statistical significance difference in what we want to present through the graph, and please reconstruct it as a way to increase readers' readability as shown in the table.
Please check and correct the above three contents.
I will re-review it after revising it.
Author Response
Reviewer 1
- Comment: The statistical method used in this study needs to be modified. If you want to compare gender and each DJ in this study, please use two-way means ANOVA. If there is an interaction, please conduct a post-examination using independent t-test for gender difference and one-way ANOVA for each DJ.
Respond: Our variables were analyzed with a two-way mixed-model ANOVA for sex and three RHs. We have added the sentence by the reviewer suggested for a clearer description. Please refer to lines 152-153.
- Comment: The tables and figures presented in this study have difficulties in interpreting the results. For the convenience of readers, please make it easier to correct the categories presented and indicate the data, statistical values, and p values. There is no indication of statistical significance difference in what we want to present through the graph, and please reconstruct it as a way to increase readers' readability as shown in the table.
Respond: Thank you for your comment. This table has been revised, had been added indication of statistical significance difference. Please refer to Table1, Table2, and Table3.

Reviewer 2 Report
Major comment.
This is a well-written manuscript reporting the sex difference in the bilateral asymmetry of GRF metrics during RHDJ. Overall, there are few methodological issues to point out. However, the main concern was interpreting female's bilateral asymmetry in GRF metrics found in RHDJs. Based on the previous reports, the authors concluded that the asymmetry in the initiation of landing contact between legs might increase the risk of a knee injury; however, as the authors addressed in the limitation part, they did not include prospective risk assessment of the same participants after RHDJ testing. I feel that the author's conclusion is an over-discussion.
To address this concern, it is necessary to explain that the bilateral asymmetry shown in the present assessment was not due to the participant's adaptation to the sporting experience but was caused by the individual's undesirable neuromuscular control. To do this, the details of the participant's sporting experience should be provided and whether the sporting experience in both sexes are controlled to be equivalent in the method part. Also, during the DJ task instructions, the participants should be asked to control their legs in the same way. Details are given in the specific comments.
Modify the in-text citations. Sometimes numbers and sometimes author's name with years. Please see the author's instructions.
Specific comments
L74  Participants
[Modification]
Please provide participants’ sport experience. My concern is whether the adaptation to the sport which requires the asymmetrical leg role (such as badminton) will affect the lower limb asymmetry during the DJ test.
L74
[Clarification] Did the author control the sports experience between genders?
L96 Task instruction
[Clarification] Did the authors instruct the participants to match the initial impact timing between legs (not drop off timing)? If the authors asked them to match the landing timing between legs and the participant showed the asymmetries in GRF metrics, these asymmetries may reflect his/her negative neuromuscular controls. If the authors did not specify the landing timing matching between legs, it is difficult to distinguish the cause of asymmetries (normal sports adaptation or failed neuromuscular control). Please provide detailed information about the instruction of DJ task.
L132 Results
[Modification]
Please provide the information of the absolute value of the countermovement jump heights (CJH) for the individual participant and the statistical summary (max, min, mean, and SD). I am interested in the absolute value of the individual CJH. If there is a significant gap in the absolute values of CJH between individuals among the same-sex group, the total physical load would be different, even for the RH testing.
Discussion
L219
[Clarification] have greater lower extremity -> have greater extremity?
First paragraph
The authors concluded that the DJ from a drop height above 160% CMJ may have a relatively high injury risk. What was the absolute value (cm) of 160% CMJ height for both genders? Was this absolute height precarious landing height for male and female participants? What was the risk criteria in height for lower limb damage? Please specify.
Third paragraph
If authors want to address the association between the functional asymmetries found in this study and the potential risk of ACL injury in female participants, the authors should incorporate the effect of sports experience and instructions of DJ task to distinguish the effect of motor adaptations and the altered neuromuscular control.
L289
[Modification] The sentence staring ‘All authors have read…’ repeated twice.
Author Response
Reviewer 2
- Comment: This is a well-written manuscript reporting the sex difference in the bilateral asymmetry of GRF metrics during RHDJ. Overall, there are few methodological issues to point out. However, the main concern was interpreting female's bilateral asymmetry in GRF metrics found in RHDJs. Based on the previous reports, the authors concluded that the asymmetry in the initiation of landing contact between legs might increase the risk of a knee injury; however, as the authors addressed in the limitation part, they did not include prospective risk assessment of the same participants after RHDJ testing. I feel that the author's conclusion is an over-discussion.
Respond: Thank you for your comment. This sentence has been revised. Please refer to lines 343.
- Comment: To address this concern, it is necessary to explain that the bilateral asymmetry shown in the present assessment was not due to the participant's adaptation to the sporting experience but was caused by the individual's undesirable neuromuscular control. To do this, the details of the participant's sporting experience should be provided and whether the sporting experience in both sexes are controlled to be equivalent in the method part.
Respond: Thank you for your comment. We have been supplementary details in-text. Please refer to lines 81.
- Comment: Also, during the DJ task instructions, the participants should be asked to control their legs in the same way. Details are given in the specific comments.
Respond: Thank you for your comment. We have been added in Method paragraph. Please refer to lines 106-109.
- Comment: Modify the in-text citations. Sometimes numbers and sometimes author's name with years. Please see the author's instructions.
Respond: Thank you for your comment. We have been modified the in-text citations.
- Comment: L74 Participants [Modification] [Clarification] Please provide participants’ sport experience. My concern is whether the adaptation to the sport which requires the asymmetrical leg role (such as badminton) will affect the lower limb asymmetry during the DJ test. Did the author control the sports experience between genders?
Respond: Thank you for your comment. We have been supplementary details in Method paragraph. Please refer to lines 81. The participant has no profession players in badminton or tennis.
- Comment: L96 Task instruction [Clarification] Did the authors instruct the participants to match the initial impact timing between legs (not drop off timing)? If the authors asked them to match the landing timing between legs and the participant showed the asymmetries in GRF metrics, these asymmetries may reflect his/her negative neuromuscular controls. If the authors did not specify the landing timing matching between legs, it is difficult to distinguish the cause of asymmetries (normal sports adaptation or failed neuromuscular control). Please provide detailed information about the instruction of DJ task.
Respond: Thank you for your comment. We have been supplementary details in Method paragraph. Please refer to lines 106-109.
- Comment: L132 Results [Modification] Please provide the information of the absolute value of the countermovement jump heights (CJH) for the individual participant and the statistical summary (max, min, mean, and SD). I am interested in the absolute value of the individual CJH. If there is a significant gap in the absolute values of CJH between individuals among the same-sex group, the total physical load would be different, even for the RH testing.
Respond: The information of the absolute value of the countermovement jump heights Please refer to lines 100-102. We agree with the reviewer’s responsibility. Therefore, we had been modified GRF normalised to body weight × RH (BW × RH-1⁄2) to minimize the variation due to individual differences in body weight and drop height (Weinhandl et al., 2017). Please refer to lines 147.
- Comment: L219 [Clarification] have greater lower extremity -> have greater extremity?
Respond: Thank you for your comment. This sentence has been revised. Please refer to lines 267.
- Comment: [First paragraph] The authors concluded that the DJ from a drop height above 160% CMJ may have a relatively high injury risk. What was the absolute value (cm) of 160% CMJ height for both genders? Was this absolute height precarious landing height for male and female participants? What was the risk criteria in height for lower limb damage? Please specify.
Respond: Thank you for your comment. This sentence has been revised, added RH160% value. Please refer to lines 101. In this study we description 160% in relative height, not absolute height. Therefore, the jump height value was related to the individual's muscle strength.
- Comment: [Third paragraph] If authors want to address the association between the functional asymmetries found in this study and the potential risk of ACL injury in female participants, the authors should incorporate the effect of sports experience and instructions of DJ task to distinguish the effect of motor adaptations and the altered neuromuscular control.
Respond: Thank you for your comment. We had been revised the details of the participant's sporting experiences description, both sexes are controlled to be equivalent. Please refer to lines 81. And we add sentence in third paragraph of discussion. Please refer to lines 310-312.
- Comment: L289 [Modification] The sentence staring ‘All authors have read…’ repeated twice.
Respond: Thank you for your comment. We had been deleted the sentence. Please refer to lines 348.

Reviewer 3 Report
To the authors,
This study evaluated the sex differences of impact forces and asymmetry during the landing phase of drop jump tasks using drop heights relative to participants' maximum jumping height. The authors demonstrated that female participants had greater asymmetry between dominant and non-dominant legs during the landing phase of drop jump tasks. These results are of interest in areas of sports and exercise sciences and would contribute to the prevention of lower limb injury in young athletes. However, this manuscript has some concerns to be addressed.
Major comments:
Participants: Please add detailed characteristics of study participants. Exercise frequency and types of sports participation in study participants should be shown. In addition, the authors should show the fitness levels of participants such as muscle strength and jumping ability levels, and it is desirable that there is no sex difference between these profiles.
Sample size: Please add explanations of how the authors determined the sample size.
Statistical power: Please add statistical power or effect size.
Minor comments:
Line 3: Please write in full spelling “I-vGRF”, not abbreviation since the word is the first appearance in the main text.
Line 38 and 45: With regard to references, please change “authors’ name” to numbers.
Line 60-64: It seems that these two sentences would lead to misunderstanding of readers. And the former sentence from line 60 is overstated. So, the reviewer recommends to revise the former sentences from line 60.
Line 98: Please add absolute values of RH100%, RH 130% and RH160%.
Line 104: Please add explanations for how to use data from three successful DJ trials in each RH task. Did the authors average them or choose based on any criteria?
Line 113: Please revise more detailed explanations for a ball-kicking test, and add at least a reference.
Tables 1 to 3 title. Please move the later sentence “values are…” to footnotes.
Footnote of table 1: please change “P < .05” to “p < .05.” In addition, please unify notation throughout the manuscript.
I hope these comments will be helpful.
Author Response
Reviewer 3
- Comment: This study evaluated the sex differences of impact forces and asymmetry during the landing phase of drop jump tasks using drop heights relative to participants' maximum jumping height. The authors demonstrated that female participants had greater asymmetry between dominant and non-dominant legs during the landing phase of drop jump tasks. These results are of interest in areas of sports and exercise sciences and would contribute to the prevention of lower limb injury in young athletes. However, this manuscript has some concerns to be addressed.
Respond: Thank you for your comment.
- Comment: Participants: Please add detailed characteristics of study participants. Exercise frequency and types of sports participation in study participants should be shown. In addition, the authors should show the fitness levels of participants such as muscle strength and jumping ability levels, and it is desirable that there is no sex difference between these profiles.
Respond: Thank you for your comment. We have been supplementary details in-text. Please refer to lines 81. In addition, the participants’ individual differences have been normalised before analysis. The muscle strength of DJ ability was normalized for CMJ drop height. And jumping ability levels was normalised to body weight × RH (BW × RH-1⁄2 ) to minimize the variation due to individual differences in body weight and drop height (Weinhandl et al., 2017). Please refer to lines 147.
- Comment: Sample size: Please add explanations of how the authors determined the sample size.
Respond: We have added explanations of how to determine the sample size. Please refer to lines161.
- Comment: Statistical power: Please add statistical power or effect size.
Respond: We have added effect size. Please refer to lines 159 and the Results.
- Comment: Please write in full spelling “I-vGRF”, not abbreviation since the word is the first appearance in the main text.
Respond: Thank you for your comment. We had been revised the sentence “I-vGRF full spelling”. Please refer to lines 34.
- Comment: Line 38 and 45: With regard to references, please change “authors’ name” to numbers.
Respond: Thank you for your comment. We had been revised the sentence. Please refer to 39-46.
- Comment: Line 60-64: It seems that these two sentences would lead to misunderstanding of readers. And the former sentence from line 60 is overstated. So, the reviewer recommends to revise the former sentences from line 60.
Respond: Thank you for your comment. We had been revised the sentence. Please refer to lines 65. The L63-64 study did not calculate the asymmetry of impact forces, also not drop jumping in different jumping height.
- Comment: Line 98: Please add absolute values of RH100%, RH130% and RH160%.
Respond: Thank you for your comment. We had been revised the sentence. Please refer to lines 100.
- Comment: Line 104: Please add explanations for how to use data from three successful DJ trials in each RH task. Did the authors average them or choose based on any criteria?
Respond: Thank you for your comment. We had been revised the sentence. Please refer to lines 106. Data were averaged across the three trials for each DJs. Please refer to lines 109.
- Comment: Line 113: Please revise more detailed explanations for a ball-kicking test, and add at least a reference.
Respond: Thank you for your comment. We had been revised the sentence. Please refer to 140.
- Comment: Tables 1 to 3 title. Please move the later sentence “values are…” to footnotes.
Respond: Thank you for your comment. We had been revised the sentence. Please refer to Tables 1 to 3.
- Comment: Footnote of table 1: please change “P < .05” to “p < .05.” In addition, please unify notation throughout the manuscript.
Respond: Thank you for your comment. We had been revised all “p < .05.”

Reviewer 4 Report
The authors investigate potential sex differences in bilateral asymmetry during landing from drop jumps. Results demonstrate greater asymmetry in females, which could be of importance for injury risk. I have several points the authors must address. I also question why the authors submitted to this particular journal. Of the MDPI journals, I would have though "Sports" would be more appropriate for this submission.
Abstract, line 14. Please provide a brief rationale for the study.
Abstract, line 17. Does this mean they landed with one leg on each force plate? If so, please make clearer.
Abstract, lines 19-22. Would be useful to include some values for the asymmetries here.
Introduction, line 34. Not a particularly scientific way of writing this sentence.
Introduction, lines 38-40. You are using two different referencing styles within the same sentence. Using a mix of different referencing styles is repeated several times throughout the paper. This is the sort of mistake my first year undergraduate students make. It is not a mistake I would expect to see in a journal submission.
Introduction, line 53. What negative effects?
Introduction, line 57. Sentence needs to be re-written as it doesn't really make sense.
Introduction, lines 61-64. You say your study is the first to look at sex differences with drop jumps from relative heights, then in the next sentence talk about several studies which have seemingly already done that. This needs rephrasing.
Methods, line 103. What was classed as a successful drop jump? Did some participants have to repeat jumps until they had 3 successful ones?
Results, line 159. Needs checking to ensure correct.
Figures 1 and 2. These need to be clearer about what they actually show. They look like simple sex differences, rather than side-to-side differentials as stated in the text and legend.
Discussion, line 215. Sentence doesn't read well.
Discussion, line 218. "jump jumping" doesn't read well.
Author Response
Reviewer 4
- Comment: The authors investigate potential sex differences in bilateral asymmetry during landing from drop jumps. Results demonstrate greater asymmetry in females, which could be of importance for injury risk. I have several points the authors must address. I also question why the authors submitted to this particular journal. Of the MDPI journals, I would have though "Sports" would be more appropriate for this submission.
Respond: Thank you for your comment. We select section was “Sport and Health”. The subsection of Sport and Health is dedicated to advancing knowledge and providing practical use of research findings on the multiple aspects affecting sport and performance at different levels and gender. The section publishes original article focusing on the athletes, the topics including biomechanics, recreational sport as health enhancing activity and injury prevention and return to play. Our topic was “Sex Disparity in Bilateral Asymmetry of Impact Forces During Height-Adjusted Drop Jumps”. It’s research about bilateral asymmetry of leg in sport performance. This topic is biomechanics analysis and gender different issue. Thus, this topic is appropriate to submit in this section. In addition, we submit special issue “Assessment of Physical Fitness and Training Effect in Individual Sports” Drop jump is often used as a plyometric exercise to improve jumping performance. Training from improper drop heights and for improper durations lead to unfavorable biomechanical changes in the lower extremities when landing, which result in reduced training effects and even lower extremity injuries. This topic is used Height-Adjusted Drop Jumps” which is Individual for athlete training. The impact force of asymmetry may provide a useful monitoring tool assessment of lower exterities injure during the drop jump task. Therefore, we believe that this study is suitable for IJERPH special issue.
- Comment: Abstract, line 14. Please provide a brief rationale for the study.
Respond: We have added the brief rationale for the study. Please refer to lines 14.
- Comment: Abstract, line 17. Does this mean they landed with one leg on each force plate? If so, please make clearer.
Respond: Thank you for your comment. We had been revised the sentence. Please refer to lines 19.
- Comment: Abstract, lines 19-22. Would be useful to include some values for the asymmetries here.
Respond: Abstract has a 200-word limit. Therefore, some values for the asymmetries were show in Results.
- Comment: Introduction, line 34. Not a particularly scientific way of writing this sentence.
Respond: We have revised sentence. Please refer to lines 35.
- Comment: Introduction, lines 38-40. You are using two different referencing styles within the same sentence. Using a mix of different referencing styles is repeated several times throughout the paper. This is the sort of mistake my first year undergraduate students make. It is not a mistake I would expect to see in a journal submission.
Respond: We have checked and corrected the format.
- Comment: Introduction, line 53. What negative effects?
Respond: We have revised sentence. Please refer to lines 57.
- Comment: Introduction, line 57. Sentence needs to be re-written as it doesn't really make sense.
Respond: We have revised sentence. Please refer to lines 60.
- Comment: Introduction, lines 61-64. You say your study is the first to look at sex differences with drop jumps from relative heights, then in the next sentence talk about several studies which have seemingly already done that. These needs rephrasing.
Respond: Lines 61-64 citation did not focus on bilateral asymmetry and difference dropping height. We have revised sentence. Please refer to lines 65.
- Comment: Methods, line 103. What was classed as a successful drop jump? Did some participants have to repeat jumps until they had 3 successful ones?
Respond: We have revised the sentence. Please refer to lines 106.
- Comment: Results, line 159. Needs checking to ensure correct.
Respond: Some modifications have be made, please refer to Table 2.
- Comment: Figures 1 and 2. These need to be clearer about what they actually show. They look like simple sex differences, rather than side-to-side differentials as stated in the text and legend.
Respond: Some modifications have be made, please refer to Figure 1. and Figure 2.
- Comment: Discussion, line 215. Sentence doesn't read well.
Respond: We have revised the original sentence to “These data suggest that a relatively high impact load of lower extremity may be linked to practices of the double-legged DJ from a height above 160% height of the countermovement jump.” Please refer to lines 263.
- Comment: Discussion, line 218. "jump jumping" doesn't read well.
Respond: We have revised the original sentence to “It is important to note that the height of countermovement jump for males is nearly 15 cm higher than that for females;” Please refer to lines 266.

Round 2
Reviewer 1 Report
Thank you for your efforts.
Author Response
Thanks for your previous comment.
Reviewer 2 Report
Strictly speaking, I’m not satisfied with some of the author’s responses. The authors merely assigned short sentences to the points raised, which were not well-considered, and did not sufficiently address the issues raised in the initial review. I was expected that my suggestion would strengthen this manuscript, but the quality of the revised version fell short of my expectations.
Please check my first-round comments as well.
Line 81; The important this was not the performance level of the participant but the type of sport they experienced. If the authors recruited the athletes who were habituated to the ipsilateral sports such as badminton, the asymmetry found in the GRF metrics may be due to their sports adaptation rather than the abnormal motor control.
Line 81; Only male participants were from Division 3?
Line 106; the participants WERE asked
Line 147;
you mean normalized to body weight * RH (BW-1*RH-1) ?
Line 310--312; How was this conclusion supported by author’s result?
In addition, the English expertise should check this manuscript.
Author Response
Strictly speaking, I’m not satisfied with some of the author’s responses. The authors merely assigned short sentences to the points raised, which were not well-considered, and did not sufficiently address the issues raised in the initial review. I was expected that my suggestion would strengthen this manuscript, but the quality of the revised version fell short of my expectations.
Please check my first-round comments as well.
- Line 81; The important this was not the performance level of the participant but the type of sport they experienced. If the authors recruited the athletes who were habituated to the ipsilateral sports such as badminton, the asymmetry found in the GRF metrics may be due to their sports adaptation rather than the abnormal motor control.
Response 1: Thanks for your comment. Males are 7 basketball, 2 track and field and 1 hockey stick players. Females are 5 basketball, 4 track and field and 1 hockey stick players. There are no habituated to the ipsilateral sports such as badminton, table tennis and tennis players. Thus, the asymmetry found in the GRF metrics not due to their sports adaptation. We added sentence. “Males are 7 basketball, 2 track and field and 1 fieldhockey athletes. Females are 5 basketball, 4 track and field and 1 field hockey stick athletes. “ Please refer line 81.
- Line 81; Only male participants were from Division 3?
Response 2: Thanks for your comment. The sentence has been altered
“10 male (1.75 ± 0.07 m, and 69.9 ± 8.9 kg) and 10 female (1.61 ± 0.04 m, and 52.4 ± 4.2 kg) 79 college athletes from the physical education department participated and their written informed 80 consent were all collected.” Please refer line 79.
- Line 106; the participants WERE asked
Response 3: Thanks for your comment. The sentence has been altered.
Original sentence:
During the DJ task instructions, the participants be asked to control their legs in the same way.
Altered sentence:
During the DJ task instructions, the participants were asked to control their legs in the same way.
- Line 147; you mean normalized to body weight * RH (BW-1*RH-1) ?
Response 4: Thanks for your comment. The sentence has been altered.
Original sentence:
normalised to body weight × RH (BW × RH-½) to minimize the variation due to individual differences in body weight and drop height
Altered sentence:
normalised to body weight × RH-½ (BW × RH-½) to minimize the variation due to individual differences in body weight and drop height.
- Line 310--312; How was this conclusion supported by author’s result?
Response 5: Thanks for your comment. The sentence has been altered “In the current study, male and female subjects used the same instructions of DJ task and had similar sports experience. Among them, athletes of the ipsilateral sports, such as badminton, table tennis, and tennis, were not included. Therefore, we infer that females had larger bilateral asymmetry, which may be due to individual's undesirable neuromuscular control.” Please refer line 253.
- In addition, the English expertise should check this manuscript.
Response 6: Thanks for your comment. we have reviewed the manuscript very carefully to improve the English and ensure that we have addressed all of your comments.

Reviewer 4 Report
I thank the authors for addressing my comments. I would suggest the opening sentences of the abstract be re-written, as they sound more like results than background to the study, though don't see the need for this to go through the review process again.
Author Response
I thank the authors for addressing my comments. I would suggest the opening sentences of the abstract be re-written, as they sound more like results than background to the study, though don't see the need for this to go through the review process again.
Respond: Thanks for your comment. The sentence has been altered
Original sentence:
Abstract: Higher risk of lower extremity injury was found for females. The biomechanical characteristics of landing were associated with the risk of injury. The manipulation of loadings should take individual differences into account.
Altered sentence:
Abstract: Side-to-side asymmetry of lower extremities may influence the risk of injury associated with drop jump. Moreover, drop heights using relative height across individuals based on respective jumping abilities could better explain lower extremity loading impact for different genders.
